# Learning Neural Acoustic Fields

**Andrew Luo**
Carnegie Mellon University

**Yilun Du**
Massachusetts Institute of Technology

**Michael J. Tarr**
Carnegie Mellon University

**Joshua B. Tenenbaum**
MIT BCS, CBMM, CSAIL

**Antonio Torralba**
Massachusetts Institute of Technology

**Chuang Gan**
UMass Amherst and MIT-IBM Watson AI Lab

## Abstract

Our environment is filled with rich and dynamic acoustic information. When we walk into a cathedral, the reverberations as much as appearance inform us of the sanctuary's wide open space. Similarly, as an object moves around us, we expect the sound emitted to also exhibit this movement. While recent advances in learned implicit functions have led to increasingly higher quality representations of the visual world, there have not been commensurate advances in learning spatial auditory representations. To address this gap, we introduce Neural Acoustic Fields (NAFs), an implicit representation that captures how sounds propagate in a physical scene. By modeling acoustic propagation in a scene as a linear time-invariant system, NAFs learn to continuously map all emitter and listener location pairs to a neural impulse response function that can then be applied to arbitrary sounds. We demonstrate NAFs on both synthetic and real data, and show that the continuous nature of NAFs enables us to render spatial acoustics for a listener at arbitrary locations. We further show that the representation learned by NAFs can help improve visual learning with sparse views. Finally we show that a representation informative of scene structure emerges during the learning of NAFs. Project site: https://www.andrew.cmu.edu/user/afluo/Neural_Acoustic_Fields

## 1   Introduction

The sound of the ball leaving the bat, as much as its visible trajectory, tells us whether the hit is likely to be a home run or not. Our experience of the world around us is rich and multimodal, depending on integrated input from multiple sensory modalities. In particular, spatial acoustic cues provide us with a sense of the direction and distance of a sound source without needing visual confirmation, allow us to estimate the properties of a surrounding environment, and are critical to subjective realism in gaming and virtual simulations.

Recent progress in implicit neural representations has enabled the construction of continuous, differentiable representations of the visual world directly from raw image observations [Sitzmann et al., 2019, Mildenhall et al., 2020, Niemeyer et al., 2020, Yariv et al., 2020]. However, our perception of the physical world is informed not only by our visual observations, but also by the spatial acoustic cues present in the environment. As a preliminary step in learning the acoustic properties of scenes, we explore an implicit model that represents the underlying impulse response of audio reverberations. As shown in Figure 1, our model can model the spatial propagation of sound in a physical scene.

Past work has explored capturing the underlying acoustics of a scene [Raghuvanshi and Snyder, 2014, 2018, Chaitanya et al., 2020]. These models, however, require handcrafted parameterizations which,

36th Conference on Neural Information Processing Systems (NeurIPS 2022).

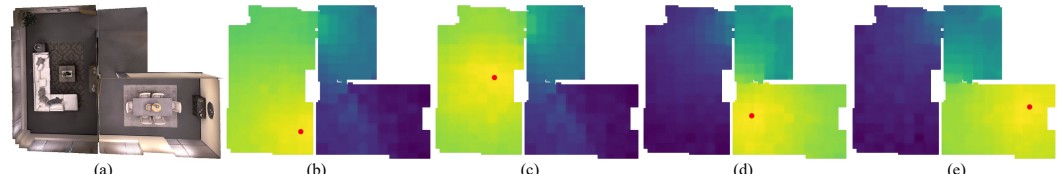

Figure 1: Neural Acoustic Field (NAF) learns an implicit representation for acoustic propagation. **(a)** A 3D top-down view of the house with two rooms. **(b)-(e)** The loudness of acoustic field as predicted by our NAF is visualized for an emitter located at the red dot. Notice how sound does not leak through walls, and the portaling effect open doorways can have. Louder regions are shown in yellow.

critically, prevent such approaches from being applied to arbitrary scenes. In this work, we extend this approach by constructing an implicit neural representation which captures, in a *generic manner*, the underlying acoustics of a scene.

Learning a representation of scene acoustics poses several challenges compared to the visual setting. First, how do we generate plausible audio impulse responses at each emitter-listener position? While we may represent the visual appearance of a scene with an underlying three-dimensional vector, an acoustic reverberation (represented as an impulse response) can consist of over 10,000 time-domain values and, thus, is significantly harder to capture. Second, how do we learn an acoustic neural representation that densely generalizes to novel emitter-listener locations? In the visual setting, ray-tracing can enforce view consistency across large portions of a visual scene (modulo occlusions). While in principle, in a similar manner, we may reflect acoustic "rays" in a scene represented as an implicit function to obtain an impulse response, an intractable amount of compute is necessary to obtain the desired representation [Srinivasan et al., 2021].

To address both challenges, we propose Neural Acoustic Fields (NAFs). To capture the complex signal representation of impulse responses in a compact and spati ally continuous fashion, NAFs encode and represent an impulse-response in the time-frequency domain. Motivated by the strong influence of nearby geometry on anisotropic reflections [Raghuvanshi and Snyder, 2018], we propose to condition NAFs on local geometric information present at both the listener and emitter locations when decoding the impulse response. In our framework, local geometric information is learned directly from impulse responses. Such a decomposition facilitates the transfer of local information captured from training emitter-listener pairs to novel combinations of emitters and listeners.

We show that NAFs are able to outperform baselines in modeling scene acoustics, and provide detailed analysis of the design choices in NAFs. We further illustrate how the structure learned by NAFs can improve cross-modal generation of novel visual views of a scene. Finally, we illustrate how the learned representation of NAFs enable the downstream application of inferring scene structure.

## 2   Related Work

**Audio Field Coding**   There is a rich history of sound field representation, encoding, and interpolation methods for 3D spatial audio. Some approaches seek to directly approximate the sound field [Mignot et al., 2013, Antonello et al., 2017, Ueno et al., 2018] while adding handcrafted priors. Others adopt a parametric representation that seek to model only the perceptual cues [Raghuvanshi and Snyder, 2014, 2018, Chaitanya et al., 2020, Mehra et al., 2014, Ratnarajah et al., 2021]. Since the complete acoustic field of a scene is computationally prohibitive to simulate in real time, and expensive to store in full fidelity, these methods have typically relied on a handcrafted encoding of the acoustic field, prioritizing efficiency above reproduction fidelity. In recent years, there has been interest in using deep learning to directly learn a sound field from data, without making strong assumptions about the scene. However in practice, these approaches use either a stationary listener or emitter [Richard et al., 2020, 2022]. In contrast, our work enables the querying of the sound field for arbitrary emitter and listener locations.

**Implicit representations**   Our approach towards modeling the underlying acoustics a scene relies on the use of a neural implicit representations. Implicit representations have emerged as a promising representation of 3D geometry  [Niemeyer et al., 2019, Chen and Zhang, 2019, Park et al., 2019, Saito et al., 2019, Hong et al., 2022] and appearance [Sitzmann et al., 2019, Mildenhall et al., 2020, Niemeyer et al., 2020, Yariv et al., 2020, Wang et al., 2021] of a scene. Compared to traditional discrete representations, implicit representations are a continuous mapping capable of capturing data

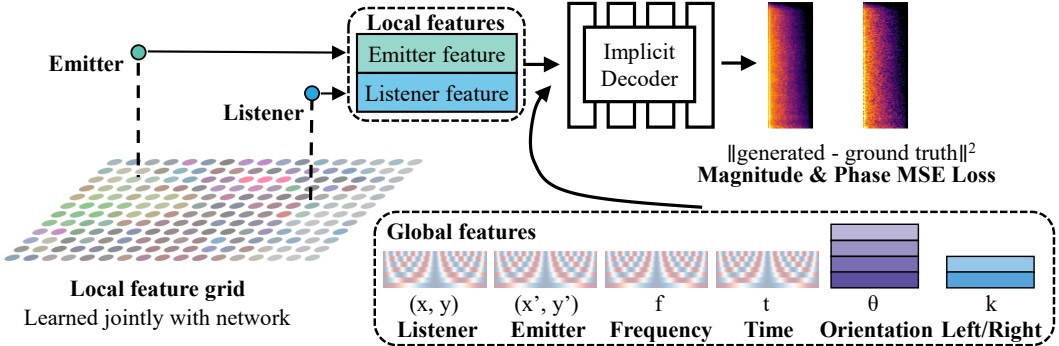

Figure 2: Overview of our NAF architecture where listener and emitter share a feature grid. Given a listener position and an emitter location, we first query a grid for local features which are learned together with the network during training. We compute the sinusoidal embedding of the positions, frequency, and time, and query a discrete embedding matrix using the orientation and left/right ear. Our method predicts magnitude and phase.

at an "infinite resolution". In [Jiang et al., 2020] proposed a grid based representation for implicit scenes, while more recently [DeVries et al., 2021] has adopted spatial conditioning for 3D image synthesis, where in both settings, the grid enables a higher-fidelity encoding of the scene. Our work also leverages local grids to model acoustics, but as an inductive bias and way to generalize to novel inputs.

**Audio-Visual Learning** Our work is also closely related to joint modeling of vision and audio. By leveraging the correspondence between vision and audio, work has been done to learn unsupervised video and audio representations [Aytar et al., 2016, Arandjelovic and Zisserman, 2017], localize objects that emit sound [Senocak et al., 2018, Zhao et al., 2018], and jointly use vision and audio for navigation [Chen et al., 2020]. Recent work aims to propose plausible reverberations or sounds from image input [Singh et al., 2021, Du et al., 2021], these approaches model the STFT using either convolution or implicit functions, which we also utilize. Different from them, our work leverages the geometric features learned by modeling acoustic fields to improve the learning of 3D view generation.

## 3 Methods

We are interested in learning a generic acoustic representation of an arbitrary scene, which can capture the underlying sound propagation of arbitrary sound sources across both seen and unseen locations in a scene. We first review relevant background information towards modeling environment reverberations. We then describe Neural Acoustic Fields (NAFs), a neural field which we show can capture, in a generic manner, the acoustics of arbitrary scenes. We further discuss how we can parameterize NAF so that it can capture acoustics property even at unseen sound sources and listener positions. Finally, we discuss the implementation details of our model illustrated in Figure 2.

### 3.1 Background on the Propagation of Sound

The sound emitted by a sound source undergoes decay, occlusion, and scattering due to both the geometric and material properties of a scene. For a fixed location pair $(\boldsymbol{q}, \boldsymbol{q}')$, we define the impulse-response at a listener position $\boldsymbol{q}$, as the sound pressure $p(t; \boldsymbol{q}, \boldsymbol{q}')$ induced by an impulse at $\boldsymbol{q}'$.

Given an accurate model of the impulse-response $p(t; \boldsymbol{q}, \boldsymbol{q}')$, we may model audio reverberation of any sound waveform $s(t)$ emitted at $\boldsymbol{q}'$, by computing the response $r(t, \boldsymbol{q}, \boldsymbol{q}')$ at listener location $\boldsymbol{q}$ by querying the continuous field and using temporal convolution:

$$r(t; \boldsymbol{q}, \boldsymbol{q}') = s(t) \circledast p(t; \boldsymbol{q}, \boldsymbol{q}') \tag{1}$$

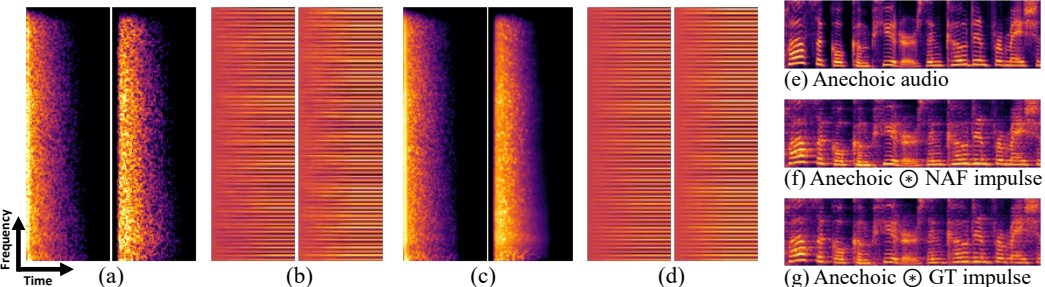

Figure 3: **Qualitative Visualization of Test Set Impulse Response Prediction.** **(a)** Ground truth log-magnitude. **(b)** Ground truth phase. **(c)** NAF predicted log-magnitude. **(d)** NAF predicted phase. **(e)** log-magnitude of an anechoic audio (without any reverberation). **(d)** The sound with a reverberation impulse response from our NAF. **(e)** The sound with the ground truth reverberation impulse response applied. Note phases are unwrapped for visualization purposes. Time & frequency are on the horizontal and vertical axes.

## 3.2   Neural Acoustic Fields

We are interested in constructing a continuous representation of the underlying acoustics of a scene, which may specify the reverberation patterns of an arbitrary sound source. The parameterization of an impulse-response introduced in Section 3.1 provides us with a method to model audio propagation when given an omnidirectional listener and emitter. To construct a model of a directional listener, we need to further model the 3D head orientation $\theta \in \mathbb{R}^2$, and ear $k \in \{0, 1\}$ (binary left or right) of a listener, in addition to the spatial position $q \in \mathbb{R}^3$ of the listener and $q' \in \mathbb{R}^3$ of the emitter.

We may then model the time domain impulse response $v$ using a neural field $\Phi$ which takes as input the listener and emitter parameters:

$$\Phi : \mathbb{R}^8 \times \{0, 1\} \to \mathbb{R}^T; (q, \theta, k, q') \to \Phi(q, \theta, k, q') = v \qquad (2)$$

Directly outputting the impulse-response waveform $v \in \mathbb{R}^T$ in the time domain with a neural network is difficult due to its high dimensional (over 10,000 elements) and chaotic nature. A naïve solution would be to further add $t$ as an additional argument to our neural field, but we found that such a solution worked poorly, due to the highly non-smooth representation of the waveform (see supplementary). We instead encode the impulse-response utilizing a short-time Fourier transform (STFT) denoted $v_{\text{STFT}}$, which we find to be significantly more amenable to neural network prediction due to the smoother nature of the time-frequency space. In Figure 3 we show magnitude spectrograms for ground truth impulse responses and those learned by our network. As $v_{\text{STFT}}$ is a complex value, we further factorize $v_{\text{STFT}}$ into log-magnitude and phase angle components. For phase angle, we use the instantaneous frequency (IF) representation proposed in GANSynth [Engel et al., 2019]. To compute the IF representation, the phase angle is unwrapped and has the finite difference taken across the time dimension for each frequency in the STFT. This transformation results in a phase representation that conducive to learning due to more regular structure.

Thus, our parameterization of NAF is a neural field $\Omega$ that is trained to estimate the impulse response function $\phi$, and outputs $[v_{\text{STFT\_mag}}, v_{\text{STFT\_IF}}]$ for a given time and frequency coordinate:

$$\Omega : \mathbb{R}^{10} \times \{0, 1\} \to \mathbb{C}$$
$$(q, \theta, k, q', t, f) \to \Omega(q, \theta, k, q', t, f) \approx [v_{\text{STFT\_mag}}(t, f), v_{\text{STFT\_IF}}(t, f)] \qquad (3)$$

We train our model using MSE loss between the generated and ground truth spectrograms $v_{\text{STFT}}$:

$$\mathcal{L}_{\text{NAF}} = \|\Omega(q, \theta, k, q', t, f)_{\text{mag}} - v_{\text{STFT\_mag}}(t, f)\|^2 +$$
$$\alpha \|\Omega(q, \theta, k, q', t, f)_{\text{IF}} - v_{\text{STFT\_IF}}(t, f)\|^2 \qquad (4)$$

across spectrogram coordinates $t$ and $f$. Where $\alpha$ is a scaling value used to balance the two losses.

## 3.3   Generalization through Local Geometric Conditioning

We are interested in parameterizing the underlying acoustic field, so that we may not only accurately represent impulse-response at emitter-listener pairs we see during training, but also at novel combina-

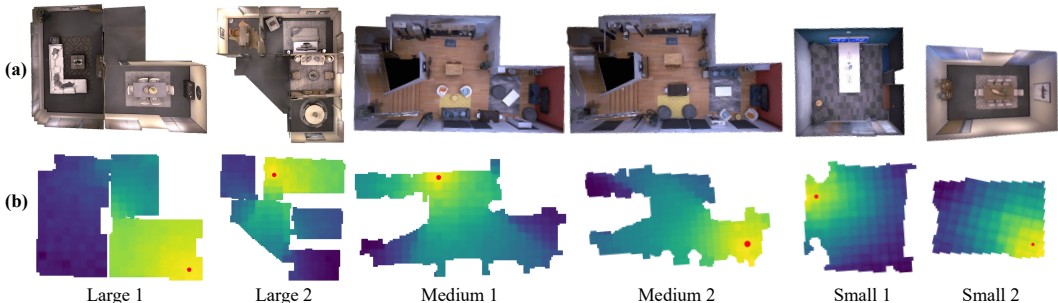

Large 1      Large 2      Medium 1      Medium 2      Small 1      Small 2

Figure 4: **Qualitative Visualization of Neural Acoustic Fields. (a)** Top down view of the rooms. **(b)** Results as inferred by our neural acooustic field. Loudness of a sound given a emitter location indicated in red, lighter color indicates louder sound. Note how openings and walls lead to portaling and occlusion of the sound.

tions of emitter and listener seen at test time. Such generalization may be problematic when directly parameterizing NAFs utilizing a MLP with inputs specified in Eqn (3), as the network may learn to directly overfit and entangle the relation between emitter and listener impulse-responses.

What generic information may we extract from a given impulse-response between an emitter and listener? In principle, extracting the full dense geometric information in a scene would enable us to robustly generalize to new emitter and listener locations. However, the amount of geometric information available in a particular impulse-response, especially for positions far away from either current emitter and listener is limited, since these positions have little impact on the underlying impulse-response. In contrast, the local geometry near either emitter and listener positions will have a strong influence in the impulse-response, as much of the anisotropic reflection comes from such geometry [Paasonen et al., 2017]. Inspired by this observation, we aim to capture and utilize local geometric information, near either emitter or listener locations, as a means to predict impulse-responses across novel combinations.

To parameterize and represent these local geometric features, we learn a 2D grid of spatial latents which we illustrate in Figure 2. The spatial latents are randomly initialized and uniformly distributed in the room. When predicting an impulse-response at a given emitter and offset position, we query the learned grid features at both emitter and listener positions, and provide it as additional context into our NAF network $\Omega$. Such features provide rich information on the impulse-response, enabling NAF to generalize better to unseen combinations of both emitter and listener locations. In the rest of this work, we refer to the NAFs with local geometric features as $\Omega_{grid}$. We learn grid latent features jointly with the underlying parameters of NAF. Additional details can be found in the supplementary.

Such a design choice, however, still requires us to consider how to further combine local geometric information captured separately from either listeners or emitters. A naïve implementation would be to maintain separate feature grids for both listener and emitter positions. Such an approach fails to account for the fact that the local geometric information captured by emitter may also inform the local geometric information around a listener. Examining Green's function, which is the solution to the wave equation, we note that it is in fact symmetric with respect to exchanging either listener or emitter positions [Chaitanya et al., 2020], indicating that the impulse-response does not change when omnidirectional emitters and listeners are swapped (acoustic reciprocity). Such a result means that we may in fact utilize the local geometric information captured near an emitter position interchangeably for either emitters and listeners. Thus, we propose our local geometric information as a single latent grid, which we show to outperform the naïve dual grid implementation.

## 4 Experiments

In this section, we demonstrate that our model can faithfully represent the acoustic impulse response at seen and unseen locations. Additional ablation studies verify the importance of utilizing local geometric features to enable test time generation fidelity. Next, we demonstrate that learning acoustic fields could facilitate improved visual representations when training images are sparse. Finally we show that the learned NAF can be used to infer scene structure.

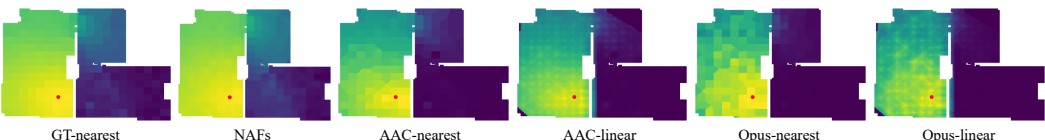

| GT-nearest | NAFs | AAC-nearest | AAC-linear | Opus-nearest | Opus-linear |

Figure 5: **Comparison of the acoustic fields.** From left to right, we visualize the loudness maps generated by the full ground truth, our NAFs, and by AAC or Opus coding combined with linear and nearest neighbor interpolation on the training set. Emitter location shown in red. Our method can faithfully reproduce the loudness map present in the ground truth.

## 4.1 Setup

For each scene, we holdout $10\%$ of the RIRs randomly as a test set. Each scene is trained for 200 epochs, which takes around 6 hours for the largest scenes on four `Nvidia V100s`. In each batch, we sample 20 impulse responses, and randomly select $2,000$ frequency & time pairs within each spectrogram. An initial learning rate of $5 \times 10^{-4}$ is used for the network and the grid features. We add a small amount of noise sampled from $\mathcal{N}(0, 0.1)$ to each coordinate during training to prevent degenerate solutions. For evaluating the learned acoustic fields, we use two different datasets:

**Soundspaces.** Soundspaces [Chen et al., 2020, Straub et al., 2019] is a synthetic dataset generated via ray-tracing, and is auralized with an ambisonic head-related transfer function (HRTF). This dataset consists of $R_i$ probe points for each scene, with each probe capable of representing an emitter or listener location for up to $R_i^2$ emitter and listener pairs. The emitters are represented as omnidirectional, while the listener acts as a stereo receiver that can have one of four different orientations. The listeners and emitters are at fixed height. Our NAFs are trained on 6 representative scenes, where 2 consist of multi-room layouts; 2 consist of a single room with a non-rectangular walls; and 2 consist of a single room with rectangular walls as in Figure 4.

**MeshRIR.** The MeshRIR dataset [Koyama et al., 2021] is recorded from a real scene, and contains monaural data collected from a cuboidal room. The listener locations are at fixed height. The emitters surround the listeners both above and below the listener plane.

## 4.2 Architecture Details

The Soundspaces dataset lacks the full parameterization of an acoustic field described in Equation 3, so we train NAF with a restricted parameterization that is available in the dataset. This allows for two degrees of freedom along the $x - y$ plane for the listener locations $q \in \mathbb{R}^2$ and the emitter location $q' \in \mathbb{R}^2$. The listener is binaural with $k \in \{0, 1\}$, and can assume four possible orientations $\theta \in \{0, 90, 180, 270\}$, while the emitter is omnidirectional. In particular, we utilize a parameterization of $\Omega_{\text{grid}}$ which maps an input tuple $[x, y, x', y', f, t] \times \{0, 90, 180, 270\} \times \{0, 1\}$ to two scalar values that represents the magnitude and phase for a given time and frequency in the STFT:

$$\Omega_{\text{grid}}(x, y, \theta, k, x', y', t, f) \Rightarrow [\boldsymbol{v}_{\text{STFT\_mag}}(t, f), \boldsymbol{v}_{\text{STFT\_IF}}(t, f)] \tag{5}$$

To encode the rotation $\theta$, as there are only 4 possible discrete rotations in the dataset, we directly query into a learnable embedding matrix of shape $\mathbb{R}^{4 \times n}$, returning a $\mathbb{R}^{1 \times n}$ vector. Similarly, to encode the left and right ear $k$, we similarly query into a learnable embedding matrix of shape $\mathbb{R}^{2 \times n}$, returning a $\mathbb{R}^{1 \times n}$ vector. The $f, t$ tuple representing the frequency and time respectively are scaled to $(-1, 1)$ and processed with sinusoidal encoding using 10 frequencies of $\sin$ and $\cos$. For MeshRIR, we set emitter $q' \in \mathbb{R}^3$ to account for the emitters that can vary in height, and do not utilize the orientation or binaural embedding.

To obtain local geometric features for either an emitter or listener in a scene, we assume that our scene is contained within a set of pixels $\mathcal{P} = \{P_1...P_k\}$ which form a grid over the scene. For a given position tuple $(x, y)$ as query location, we then interpolate the local features. Where $\mathcal{L}(\cdot)$ is the interpolation function. $(p_1^* ... p_k^*)$ are the set of all pixel that form the grid, and $\tilde{f}(\cdot)$ represents the features stored at a given pixel:

$$(x, y) \Rightarrow \mathcal{L}(x, y; \tilde{f}(p_1^*), \dots \tilde{f}(p_k^*)) \tag{6}$$

$$= \sum_{i=1}^{k} w_i \tilde{f}(p_i^*) \tag{7}$$

$w_i$ is determined by a Nadaraya-Watson estimator with a Gaussian weighting kernel applied to the distance between query and grid coordinates:

$$w_i = K((x, y), (x_i, y_i)) / \sum_{j=1}^{k} K((x, y), (x_j, y_j)) \tag{8}$$

$$K(\boldsymbol{x}, \boldsymbol{x}') = \exp(-\|\boldsymbol{x} - \boldsymbol{x}'\|_2^2 / 2\sigma^2) \tag{9}$$

Because this interpolation function is differentiable, we jointly learn the grid features during training. These queried features are combined with the coordinates processed with sinusoidal encoding using 10 frequencies of sin and cos functions. We process both the listener and emitter position tuples this way. We combine the grid based features with the sinusoidal embeddings and the discrete indexed embeddings as the input to our multilayer perceptron $f_\phi$. Please refer to Figure 2 for a visualization of our model, and supplementary for further details. We compare using a shared local geometric feature with the emitter and listener, as well as having the emitter and listener query their own individual grids.

## 4.3 Evaluation of Neural Acoustic Fields

We first validate that we can capture environmental acoustics at unseen emitter-listener positions.

**Baselines.** We compare our model against two widely used high performance audio coding methods: Advanced Audio Coding (AAC) and Xiph Opus. We use low bitrates in order to attempt to approach the storage costs for our NAF. For each method, we apply both linear and nearest neighbor interpolation to the coded acoustic fields. Both linear and nearest neighbor approaches are widely used [Savioja et al., 1999, Raghuvanshi et al., 2010, Pörschmann et al., 2020] in modeling of spatial audio. We further implement the binaural DSP baseline described by [Richard et al., 2020], which uses the image source method and a KEMAR HRTF. We also compare the listener and emitter either sharing or using individual local geometric features in our NAFs.

| Method | Storage (MiB) |
|--------|---------------|
| AAC | 312.07 |
| Opus | 163.23 |
| NAF (Shared) | 8.41 |

Table 1: **Average space consumption across 6 Soundspaces scenes. Lower is better.**

Each method is provided with the same train-test split. We visualize the acoustic fields produced by different methods in Figure 5. Details of our baselines can be found in the supplementary section G.

**Metrics.** We evaluate the results of our synthesis by measuring the spectral loss [Défossez et al., 2018] between the generated and the ground truth log-spectrograms, as well as measuring the percentage error between the T60 reverberation time in the time domain. In this case, lower spectral loss and T60-error values indicates a better result. Additional quantitative results can be found in the supplementary. We also perform a human evaluation where subjects are presented with a two-alternative forced-choice task. Each trial requires selecting if the NAFs or the AAC-nearest auralized music samples best match with ground truth auralization.

**Results.** As shown in Table 2, our NAFs achieve significantly higher quality on the modeling of unseen impulse responses compared to strong interpolation baselines across all six scenes. Observing the qualitative results in Figure 4, we observe that NAFs can predict smoothly varying acoustic fields that are affected by the physical surroundings. Extending our model to MeshRIR which is captured from a real scene, we observe that our NAFs continue to perform better on both spectral and T60 metrics. Comparing our results against baselines in Figure 5 and Table 1, our methods are able to better approximate the ground truth at a fraction of the storage cost, and does not exhibit the sound energy leakage present in linear interpolation. The size of a spatial acoustic field is important for real life applications. In our human evaluation with 21 subjects who were asked to judge 10 test-time RIRs, 82.38% of responses indicates that our NAFs were higher quality compared to AAC-Nearest.

| Model | Large 1 Spec.↓ | Large 1 T60↓ | Large 2 Spec.↓ | Large 2 T60↓ | Medium 1 Spec.↓ | Medium 1 T60↓ | Medium 2 Spec.↓ | Medium 2 T60↓ | Small 1 Spec.↓ | Small 1 T60↓ | Small 2 Spec.↓ | Small 2 T60↓ | MeshRIR Spec.↓ | MeshRIR T60↓ | Mean Spec.↓ | Mean T60↓ |
|---|---|---|---|---|---|---|---|---|---|---|---|---|---|---|---|---|
| AAC-nearest | 1.913 | 9.996 | 1.989 | 13.31 | 2.111 | 6.148 | 2.122 | 6.051 | 2.296 | 9.798 | 2.509 | 5.809 | 1.057 | 4.740 | 1.999 | 7.979 |
| AAC-linear | 1.904 | 8.847 | 1.964 | 11.63 | 2.105 | 4.585 | 2.116 | 4.422 | 2.299 | 8.253 | 2.521 | 6.021 | 1.081 | 6.697 | 1.998 | 7.208 |
| Opus-nearest | 1.740 | 12.20 | 1.817 | 15.15 | 1.887 | 7.875 | 1.898 | 7.897 | 2.058 | 10.68 | 2.238 | 7.564 | 1.711 | 5.068 | 1.907 | 9.493 |
| Opus-linear | 1.780 | 11.30 | 1.827 | 13.55 | 1.922 | 6.710 | 1.934 | 6.917 | 2.097 | 9.116 | 2.284 | 6.981 | 1.743 | 5.768 | 1.941 | 8.621 |
| DSP | 1.106 | 14.62 | 1.170 | 13.68 | 1.064 | 10.24 | 1.067 | 9.732 | 1.079 | 12.77 | 1.097 | 11.03 | N/A | N/A | 1.097 | 12.01 |
| NAF (Dual) | 0.413 | 6.288 | 0.421 | 7.111 | 0.386 | 3.173 | 0.387 | 3.169 | 0.365 | 3.497 | 0.361 | 2.210 | **0.403** | 4.201 | 0.388 | 4.241 |
| NAF (Shared) | **0.396** | **4.166** | 0.413 | **6.075** | **0.384** | **3.110** | 0.384 | **3.072** | 0.356 | **3.378** | 0.344 | **2.098** | 0.403 | **4.191** | **0.380** | **3.650** |

Table 2: **Quantitative Results on Test Set Accuracy.** We report the spectral loss between generated and ground truth log spectrograms across methods, as well as the percentage (%) difference for the T60 reverberation time. The best method for each room is **bolded**. For the nearest and linear baselines, we perform interpolation in the time domain using samples from the training set. The DSP is not implemented for MeshRIR due to the lack of absolute room coordinates.

| | Large Room 1 | | | | | | Large Room 2 | | | | | |
|---|---|---|---|---|---|---|---|---|---|---|---|---|
| | PSNR ↑ | | | SSIM ↑ | | | PSNR ↑ | | | SSIM ↑ | | |
| Training Images | 75 | 100 | 150 | 75 | 100 | 150 | 75 | 100 | 150 | 75 | 100 | 150 |
| NeRF | 25.41 | 27.36 | 29.85 | 0.872 | 0.892 | 0.926 | 25.70 | 27.74 | 29.34 | 0.821 | 0.853 | 0.879 |
| NeRF + NAF | **26.19** | **27.59** | **29.90** | **0.895** | **0.911** | **0.927** | **26.24** | **28.22** | **29.45** | **0.837** | **0.866** | 0.879 |

Table 3: **Quantitative Results on Cross-Modal Image Learning.** Quantitative results on joint training of NeRF and NAF jointly conditioned on a single local grid. We use very sparse training images in highly complex scenes. When evaluated on 50 test images, we observe that cross-modal learning helps improve PSNR and SSIM when the visual training data is more sparse.

A comparison of using shared and dual local geometric features indicates that despite having fewer learnable parameters, we achieve better performance by sharing the local geometric features. Examples of individual impulse responses generated by our model are shown in Figure 3.

**Generalization through Geometric Conditioning.** We next assess the impact of utilizing local geometric conditioning as a means to generalize to novel combinations of emitter-listener positions. On the "Large 1" room, in Figure 7 we evaluate test set spectral error when NAF is trained with a limited percentage of the training data either with or without local geometric conditioning. We find that such geometric conditioning enables better test set reconstruction error, with the performance gap increasing with less data.

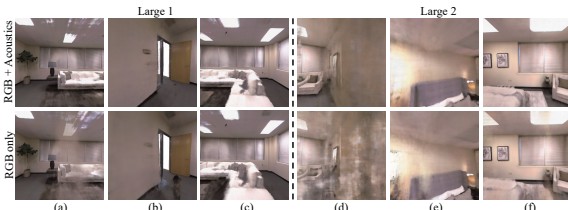

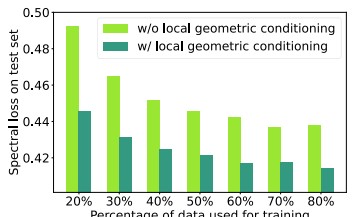

Figure 6: **Qualitative Visualization of Cross-Modal Image Learning.** Qualitative comparison between NeRF+NAFs with RGB and acoustic supervision, and NeRF learned with only RGB supervision. **(a)-(c)** Three views from "Large 1". **(d)-(f)** Three views from "Large 2".

Figure 7: **Local Geometric Conditioning.** Comparison of NAF with and without local geometric conditioning trained with different amounts of data.

## 4.4 Cross-modal learning

In this experiment, we explore the effect of jointly learning acoustics and visual information when we are given sparse visual information. Recall that our NAF includes a local geometric feature grid $\mathcal{P}$ that covers the entire scene. For our cross-modal learning experiment, we jointly learn this feature grid with a NeRF network modified to accept local features sampled from this grid along with the traditional sinusoidal embedding. In the acoustics branch, we query the grid using emitter and listener positions. In the NeRF branch, we use point samples along the ray projected on the grid plane to query the features. In both cases, the process is fully differentiable. We use a standard implementation of NeRF with a coarse and fine network.

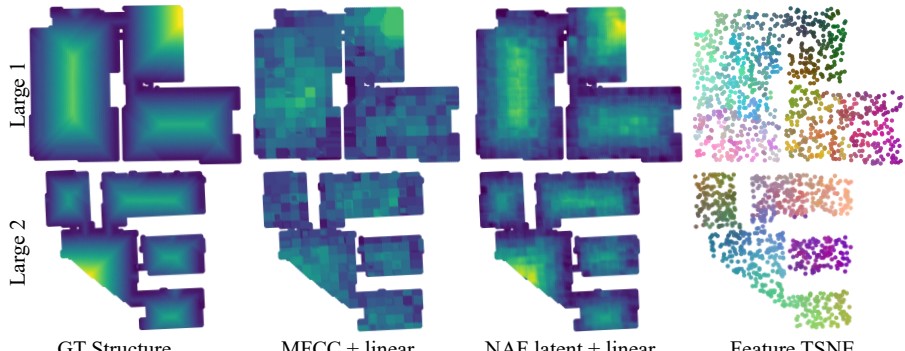

|  |  |  |  |
|:---:|:---:|:---:|:---:|
| GT Structure | MFCC + linear | NAF latent + linear | Feature TSNE |

Figure 8: **Visualization of scene structure decoding with a linear layer. Column 1:** The ground truth scene structure map, at each position we visualize the distance to the nearest wall. **2:** Linear decoding results using MFCC features. **3:** Linear decoding results using NAF features. **4:** TSNE applied to the NAF features.

In the NeRF only setting, we minimize color $C$ reconstruction loss for a ray $r$ over a batch of rays $\mathcal{R}$: $\mathcal{L}_{\text{RGB}} = \sum_{r \in \mathcal{R}} ||\hat{C}(r) - C(r)||_2^2$. In contrast, in the NAF + NeRF experiment, we jointly minimize $\mathcal{L}_{\text{RGB}} + \mathcal{L}_{\text{NAF}}$, where $\mathcal{L}_{\text{NAF}}$ is defined in equation 4. We utilize 64 coarse samples and 128 fine samples for each ray, and sample 1024 rays per batch.

**Results.** We train on the two large rooms in our training set. For each room $75, 100, 150$ images are used for training, while the same $50$ images of novel views are used for all configurations during testing. In Table 3 we observe that training with acoustic information helps improve the PSNR and SSIM of the visual output. This effect is more significant when the training images are very sparse, the NAF network helps less when there is sufficient visual information. Qualitative results are shown in Figure 6, we see there is a reduction of floaters in free space.

### 4.5 Inferring scene structure

Given a reverberant sound, humans are able to build a mental representation of the surrounding room and make a judgement about the distance of nearby obstacles Kolarik et al. [2016]. We investigate the intermediate representations constructed by our neural network in the process of learning an acoustic field, and examine if these representations can be used to decode the scene structure.

**Setup.** The intermediate representation of the NAF depends on both listener locations $q$ and emitter location $q'$, the rotation angle $\theta$, the ear $k$, the time $t$ and frequency $f$. For consistency, at a given location $(x^*, y^*)$ in the scene, we extract the NAF latent by setting the emitter location $q'_i = (x^*, y^*)_i$. For the listener location, we iterate over five randomly selected points in the scene $q \in [q_1, \ldots, q_5]$, which we keep constant for all $q'_i$. The rotation angle is fixed to $\theta = 0$, and we compute the representation average over all possible $(k, t, f)$, and concatenate latents for the selected $q$. For our NAFs, latents are extracted from the last layer prior to the output which includes 512 neurons. As a

| | Explained variance | | | | | | |
|---|---|---|---|---|---|---|---|
| **Features** | Large 1 | Large 2 | Medium 1 | Medium 2 | Small 1 | Small 2 | Mean |
| MFCC | 0.501 | 0.458 | 0.614 | 0.642 | 0.820 | 0.723 | 0.626 |
| NAF latents | **0.908** | **0.891** | **0.900** | **0.923** | **0.936** | **0.916** | **0.913** |

Table 4: **Quantitative Results on scene structure decoding.** We measure the explained variance scores of the predicted wall distance against the ground truth wall distance at test time locations after linear decoding. NAF latents consistently achieve higher explained variance scores than MFCC features.

comparison to our learned representation, we extract Mel-frequency cepstral coefficients (MFCCs) from the ground truth impulse response provided by a nearest neighbor interpolator. We use a similar setup as above, for a given location we set this to be $q'_i$, and iterate over the same five listener locations

$q_{1...5}$. We average the MFCCs over the left and right ear, and concatenate for the selected $q$. After flattening, the MFCC features are approximately 500 dimensional for any given room.

We fit a single linear layer to NAF and MFCC features respectively. For testing and visualization of the linear decoding results, we sample a regular grid of points with $0.1m$ distance between each point. For fitting the linear decoder, we randomly sample points within the scene such that the number of training points are $10\%$ as many as the testing points. For each location in the scene, we extract the distance to the nearest wall as the decoding target.

**Results.** We visualize the results of our linear decoding in Figure 8. The intermediate representation of our NAFs reveals an underlying structure that is both smooth and semantically meaningful. In the multiroom scenes, the latent is well separated for each room. We are able to successfully decode the scene structure with a linear layer when using our NAFs, but decoding fails when using MFCC features. In Table 4, we show the amount of explained variance of our decoding results on the test set. Our learned features are able to consistently achieve much higher scores than those using MFCC features.

## 5   Discussion

**Limitations and Future Work.**   Although our method achieves generalization and high quality representations of acoustic fields within a single scene, NAFs do not currently generalize to multiple scenes. Future work may explore generalization to novel scenes. One possible approaches may be to incorporate multi-modal inputs with the goal of synthesizing an acoustic field with few-shot visual or acoustic input.

**Societal impact.**   Our work focuses on learning a high quality representation of acoustic fields. The primary use case for our work lies in virtual reality and gaming. As our work can lead to more believable and higher quality representations of spatial audio than alternative methods, it is possible that our work could increase the dependency and time spent on gaming. The more compact nature of our acoustic representations may allow for spatial audio to be deployed to more systems, and enable more equitable access.

**Conclusion.**   In summary, this paper introduces Neural Acoustic Fields (NAFs), a compact, continuous, and differentiable acoustic representation which can represent the underlying reverberation of different audio sources in a scene. By conditioning NAFs locally on the underlying scene geometry, we demonstrate that our approach enables the prediction of plausible environmental reverberations even at unseen locations in the scene. Furthermore, we demonstrate that the acoustic representations learned through NAFs are powerful, and may be utilized to facilitate audio-visual cross-modal learning, as well as to infer the structure of scenes.

## Acknowledgments and Disclosure of Funding

We would like to thank Leila Wehbe for providing feedback on our paper. This work was supported by the MIT-IBM Watson AI Lab, Amazon Research Award, ONR MURI, DARPA Machine Common Sense program, ONR (N00014-18-1-2847), NIDCD R01 DC015988, Mitsubishi Electric, and the Tianqiao and Chrissy Chen Institute.

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
