# OpenReview forum: "Learning Neural Acoustic Fields"
_NeurIPS.cc/2022/Conference — NeurIPS 2022 Accept_

### Official Review · Reviewer_ossn · 2022-07-04

**Rating:** 8
**Confidence:** 4
**Soundness:** 4 excellent
**Presentation:** 4 excellent
**Contribution:** 4 excellent

**Summary:**

This paper proposed a model for learning to interpolate room impulse responses given room geometry and emitter/listening position.
Experiments on two datasets demonstrate that the proposed model can encode room geometry information, and that the learned representation can assist in cross-modal learning.

**Questions:**

Minor comment: Figure 3 shows spectrograms both upside-down (e-g) and sideways (a-d) and without axis labels.  Please, for the sake of consistency, stick to standard orientations for this kind of information.  If you're cramped for space, I suggest to simplify to a monophonic visualization as there's no information content in the left/right channels that's important for this particular figure.

**Limitations:**

There is no mention of societal impact.  The checklist response claims this will be included in a supplementary document.

Technical limitations are addressed.

**Strengths And Weaknesses:**

Strengths:

The paper is clear and well-written.  The method appears to work well.


Weaknesses:


The quantitative evaluations (eg Table 1) look good, but there is no meaningful qualitative evaluation of the method's ability to interpolate RIRs.
The raw numbers are difficult to interpret, even for knowledgable practitioners, let alone the broader machine learning audience.

---

> ### Author Response · Authors · 2022-08-02
> **Response to ossn**
>
> We are grateful to Reviewer ossn for the suggestions and comments. We address specific comments below.
>
> > **Q1) Quantitative and qualitative metrics**
>
> As part of the revision, we have additionally provided direct-to-reverberant ratio (DRR) error and interaural cross-correlation (IACC) coefficients error. The former should reflect how well we model the direct sound, while the latter should reflect binaural spatialization. In addition, we performed a human evaluation where subjects were provided with headphones and asked to perform a two-alternative forced-choice task, where over 82.38% found our NAFs to outperform the AAC-nearest baseline. We also provide qualitative samples on our project site: https://sites.google.com/view/nafs-neurips2022
>
> > **Q2) Visualization of the spectrograms**
>
> All our spectrograms are presented with frequency on the vertical axis, and time on the horizontal axis. In Figure 3., (e)-(g) show the spectrogram of a long music sample that has been convolved. We have added axis labels and adjusted the orientation of our figure to improve clarity in the revision.
>
> > **Q3) Societal impact**
>
> Due to space constraints our societal impacts section was put on the last page of our supplemental. We will add a note in our revision to indicate where this section can be found.
>
> To further clarify, the primary use case for our work lies in virtual reality and gaming. As our work can lead to more believable and higher quality representations of spatial audio than alternative methods, it is possible that our work could increase the dependency and time spent on gaming.
>
>
> Thank you for your comments! We will address your feedback in the revision.

---

### Official Review · Reviewer_e4mi · 2022-07-11

**Rating:** 8
**Confidence:** 4
**Soundness:** 4 excellent
**Presentation:** 4 excellent
**Contribution:** 4 excellent

**Summary:**

This paper describes a NeRF-like representation for binaural room impulse responses in a given acoustic space. It is an implicit representation of listener position, orientation, and ear, as well as source position, time, and frequency that outputs a scalar magnitude and phase. Experiments on the Soundspaces and MeshRIR datasets show that it is able to reproduce realistic effects in loudness maps like sonic shadows from doorways for irregularly-shaped rooms that would otherwise require expensive ray tracing for impulse response simulation. The model's predictions are compared to those of Opus and AAC and shown to be both smaller in space as well as more accurate in terms of T60 estimation and log spectrogram MSE. An ablation shows that a geometric encoding layer that is shared between listener and emitter works better than having separate or no encoding layer. Use of the proposed representation with NeRF for multi-modal representation improves the visual representation when training images are sparse. A further experiment shows that the distance to the nearest wall can be predicted more accurately from the learned representation than from MFCCs by a linear layer.

**Questions:**

What is the architecture of the actual model that is being fit? What are its hyperparameters?

Are the AAC and Opus baselines encoding just the impulse responses? Since they are designed to encode natural sounds and not impulse responses, does encoding a sound convolved with the impulse responses give more accurate results?

In listening to the videos in the supplementary material, I didn't get a great sense of direction of arrival of the sound source. Why might this be? Can this be addressed?

**Limitations:**

The limitations section captures salient limitations of the approach and experiments.

**Strengths And Weaknesses:**

Strengths:
* Novelty and timeliness: NeRF is very popular and while several papers have conditioned a visual NeRF model on audio of various kinds, this is the first, to my knowledge, that provides a meaningful application of the same idea of implicit scene encoding in an audio context.
* High quality experiments: The experiments are well conducted and thorough. Useful ablations were conducted which provide non-trivial insights, i.e., that a shared geometric encoding works better than having separate encodings for source and listener.
* Clarity: The paper is very well written and easy to follow
* Supplementary material: The supplementary material is useful and informative without being overwhelming.

Weaknesses:
* A few missing details: the paper makes very efficient use of its space, but seems to leave out a few crucial details: the actual architecture of the model being trained, how exactly AAC and Opus were used to encode these libraries of impulse responses, and how the loudness maps are calculated.

---

> ### Author Response · Authors · 2022-08-02
> **Response to e4mi**
>
> We thank Reviewer e4mi for the helpful and constructive review. We address specific questions below, and will include additional details in a revision.
>
> > **Q1) Network details**
>
> The feature grid contains 64 features at each location, and is initialized from the gaussian distribution. In the case where individual grids are used for the emitter and listener, two grids are initialized independently. The network consists of 8 fully connected layers, and leaky ReLU with a slope of 0.1 is used as the activation function. The network has two output neurons, representing log-magnitude and instantaneous frequency (phase). Each fully connected layer uses 512 intermediate feature maps. The network is trained using the Adam optimizer with an initial learning rate of 5e-4, which decays to 5e-5 at the end of the training. The code definition for the network is provided as part of our supplementary. We will update our supplementary to better detail our hyperparameters and setup.
>
> > **Q2) Baseline and visualization details**
>
> The impulse responses are indeed processed directly using the baseline encoders. This choice was motivated by our desire to have a set of impulse responses that could be applied to arbitrary sounds. It would be possible to encode the post-convolution audio, however that would sacrifice the ability to generalize. The specific code we used to encode our data is provided in the `baselines` folder of the supplementary code. For the loudness visualization we compute the root mean square of the impulse. Code for the visualization can be found in `testing/vis_loudness_NAF.py`.
>
> > **Q3) Directional sounds**
>
> For our qualitative demos, many of the instances have the emitter placed quite far away from the listener. In cases where the emitter is not in the same room as the listener, the reverberation of the sound is more obvious, while the directional nature of the sound is less so. In cases where the listener is immediately outside of the doorway, the directional aspect should be most evident (eg. Emitter location 1 in Large 2 at around 0:24; Emitter location 2 in Large 2 at around 0:10; Emitter location 1 in Large 1 at around 0:32). The use of headphones may better highlight the directional effect. We agree that losses explicitly designed for maintaining directional cues are worth exploring.
>
> We thank you for your comments, and we hope that this clarifies our results! We will update the paper to reflect your suggestions.

---

> > ### Comment · Reviewer_e4mi · 2022-08-09
> > **Response**
> >
> > I thank the authors for their detailed responses and the additional analyses of their results. I do agree with reviewer u3Tz that an evaluation of the phase of the reconstructions was missing in the original submission, but believe that the authors have sufficiently addressed this shortcoming with their analysis of the IACC error. Regarding the authors' responses to my review, thanks for providing them. In addition to being visible in the code, the baselines should be described sufficiently in the text of the main paper that a reader should be able to understand what is going on. Reading the code shouldn't be necessary to understand the experiments.

---

### Official Review · Reviewer_u3Tz · 2022-07-12

**Rating:** 8
**Confidence:** 5
**Soundness:** 3 good
**Presentation:** 3 good
**Contribution:** 4 excellent

**Summary:**

Disclosure: I reviewed a previous version of this paper submitted to another conference earlier this year.

This paper introduces the concept of Neural Acoustic Fields, inspired by the recent developments in Neural Radiance Fields (NeRF) in the context of visual scene generations from a few still images. The idea is, for a given complex acoustic environment, to learn using a neural network a function that maps a pair of 3D (emitter, receiver) locations, a receiver orientation, a receiver channel (left or right) and a time-frequency point (t,f) to the corresponding log-magnitude and phase spectrogram of the room impulse response (RIR) from that emitter to that receiver at that (t,f) point. The function is learned using a balanced least square loss in the log-magnitude and phase domains and positions are represented by a sinusoidal embedding as well as learned geometrical feature maps.



**Questions:**

I do not have specific questions except those arising from the lack of details provided in the paper (see below).

**Limitations:**

## Phase and binaural rendering
The main limitation of the paper is the evaluation. Indeed, the only two objective metrics used to evaluate generate RIRs are insufficient to validate the central claim made by the authors in the abstract, namely: "continuous nature of NAFs enables us to *render spatial acoustics* for a listener at arbitrary locations".

The first metric is the RT60 error, which is a single scalar parameters that globally captures the reverberation time but ignores fine properties of the reverberation which typically vary across frequency. The second metric used is the mean squared error on the log-magnitude spectrogram of the RIR. This metric completely ignores the phase reconstruction. Though, correctly reconstructing the phase is known to be of crucial importance for rendering reverberation, and even more so for rendering spatial effects, in binaural settings. Indeed, the primary cue for sound source localization are phase differences across frequencies, which need to be very accurately rendered. The authors claim that their model is able to render multiple orientations of a binaural listener. However, the main notable effect of rotating a listener on the spot for a fixed source in a fixed environment will be on the phase differences between the two channels. The authors to propose to predict the phase via an instantaneous frequency representation, but the reconstructed phases and phase differences are never evaluation.

In short, the evaluation is currently insufficient to support the implicit strong claim that the proposed method is able to "render spatial acoustics" in the binaural settings. They merely convince that the method is able to accurately render *monaural* reverberation, which is a significantly easier (though highly non-trivial) task. This strong limitation is not properly addressed anywhere in the manuscript.

## Clarity and precision
The writing is unprecise and handwavy at a number of places. It gives the general impression that the main background of the authors is in NeRF rather than in acoustics/audio signal processing.
- L44: "an intractable number of rays are necessary"  The term "intractable" is not precise enough here. Ray tracing is a widely used technique for acoustic simulation, for instance it was used to generate the SoundScape dataset [25] used by the authors themselves for training their method. In that paper, 200 rays are emitted from each source location. Is that considered an intractable number?
- L47 "NAFs encode and represent an impulse-response in the Fourier frequency domain." -> actually, in the time-frequency domain, which is quite different.
- L60: "The first approach encodes the sound field at a user-centric location by capturing the sound from spatially distributed sources [8,9,10,11]". It is not clear what the authors mean here by "capturing the sound from spatially distributed sources": capturing in what sense? How do those methods specifically differ from the proposed approach? Curiously, the 4 references given seem completely unrelated to each other and span 50 years of literature. The most recent one [11], from 2021, actually seems highly related to what the authors propose, and some more in-depth comparison with this work should be made in the paper.
- L113 "Directly outputting the impulse-response v magnitude" -> this seems incorrect, the term "magnitude" should probably be removed.
- L119 "due to the smooth nature of the frequency space" -> the authors probably mean "time-frequency space".
- L122 "For phase, we use the instantaneous frequency (IF) representation proposed in [29]." : more details on the phase representation used should be provided to make the paper self-contained / reproducible. In particular, one cannot directly reconstruct the phase from the instantaneous frequency, since it is by definition the time derivative of the phase. Hence, a global phase is missing at each frequency. It is not clear how this is handled by the authors.
- L147-150 on local geometric features: It is hard to understand what the authors are talking about here without reading in details the supplementary material. Would be good to make this part more self-explanatory.
- The authors compare an approach using a "shared" feature grid vs. a "dual" feature grid, the former outperforming the latter. However, in the supplementary material (fig. A4), only the network architecture for a dual grid is presented, which is confusing.
- Section 4.1: when presenting the two datasets, the authors omit to discuss their limitations, although both are in fact quite limited and do not allow testing the proposed approach in its full generality. The first one only employs 2D grids while the second one is monaural.
- L217: "The bitrates are chosen to approach the storage costs for our NAF": This is not precise enough to understand how the baselines were implemented exactly. The authors should report results as a function of bitrates for the different methods to get a clearer picture of the benefit of the approach compared to standard techniques. Moreover, Table 3 seems to contradict this statement: here we see that the space consumption of NAF is vastly inferior to that of AAC and Opus.

## Typos
- L44: "an intractable number of rays are necessary to" -> is necessary.
- L179 "on 6 representative scenes. Where" -> remove full stop
- L212 "as well as using have" -> incorrect phrasing.


**Strengths And Weaknesses:**

## Strength
- The proposed approach is sound both in terms of the underlying acoustical model and in terms of the neural network design and training process, and the general idea of NAFs is novel, interesting, and potentially impactful.
- The ability of the method to correctly estimate log-magnitude time-frequency responses and reverberation times at any point in different complex environments across a synthetic and a real dataset is convincingly demonstrated.
- A number of interesting experiments reveal that the proposed model learns geometrical structure that could be used for other purposes.

## Weakness
- An important weakness of the paper is the evaluation (See "limitations" for more details).
- The paper lacks clarity as well as many important details to make it self-contained and reproducible (See "limitations" for more details).

---

> ### Author Response · Authors · 2022-08-02
> **Response to u3Tz [1/2]**
>
> We appreciate your assessment that the NAFs are a novel and interesting idea. We thank Reviewer u3Tz for the helpful review. Below are our responses to specific comments.
>
>
> > **Q1) Evaluation on binaural/spatial rendering**
>
> We agree that binaural cues are important and should be reflected in our evaluations. The interaural cross correlation coefficient (IACC) is a commonly accepted metric for the spatial localization of sound sources from binaural audio [1], and is believed to be predictive of human localization of sound sources [2]. The IACC coefficient is computed for each binaural impulse response, and the mean absolute difference between our predicted and ground truth IACC is taken.
>
>
> |              | Large 1   | Large 2   | Medium 1  | Medium 2  | Small 1   | Small 2   | Mean      |
> |--------------|-----------|-----------|-----------|-----------|-----------|-----------|-----------|
> | AAC-nearest  | 236.8     | 184.2     | 213.7     | 215.3     | 264.8     | 272.5     | 231.2     |
> | AAC-linear   | 212.3     | 156.7     | 185.9     | 187.8     | 245.2     | 265.2     | 208.8     |
> | Opus-nearest | 73.75     | 45.97     | 71.97     | 74.70     | 103.8     | **67.40** | 72.93     |
> | Opus-linear  | 75.56     | 48.32     | 73.38     | 77.33     | 109.2     | 78.10     | 76.98     |
> | DSP          | 460.5     | 446.0     | 430.0     | 430.1     | 443.6     | 446.3     | 442.7     |
> | NAF (Dual)   | 74.01     | 45.94     | 71.89     | 74.70     | 103.8     | **67.40** | 72.96     |
> | NAF (Shared) | **73.68** | **45.90** | **71.52** | **73.58** | **103.6** | **67.40** | **72.62** |
> * Table 1. Mean absolute difference of IACC (unit in seconds, values here multiplied by 1e6). Lower is better.
>
> Our method has the lowest IACC error, which indicates that our method is capable of rendering spatial audio. We include this important metric in our revised paper. Thank you for your valuable suggestions.
>
> > **Q2) Technical clarifications**
>
> * **[Cost of ray tracing]** Soundspaces does not use 200 rays, but instead uses [5000 rays \* 200 bounces] for each listener, and [200 rays \* 10 bounces] for each emitter. We should clarify that in our paper we mean ray tracing in the context of a learned implicit neural representation of scene structure. Due to the computational cost, current state-of-the-art work in ray tracing in implicit neural representations is limited to a single bounce [3]. We will clarify this in the revision.
>
> * **[Discussion of prior work]** We agree that [4] is an important work in modeling binaural audio. However, the approach of our model and [4] are different. While [4] seeks to output the binaural audio directly, we output an impulse response which can be applied to mono audio. Secondly, NAFs model the STFT (log-magnitude and instantaneous frequency of phase), whereas [4] learns the time domain waveform. Finally, in practice [4] is trained and evaluated on data where the listener is fixed and only the emitter can move, while the NAF model is trained and evaluated on listener and emitter pairs which can both move. This requires the modeling of a much larger set of impulse responses. We did attempt to adapt [4] to our task by using an impulse function as input, and the impulse response as supervision. We could not successfully learn the impulse response in this modified setup, probably because their network was not tuned for this task. We will include additional discussion of [4] in our revision.
> * **[Instantaneous frequency]** We use the STFT phase instantaneous frequency representation proposed in GANSynth [5], which retains the phase for each frequency band. After the STFT for a waveform is computed, the phase angle within each frequency band is extracted and unwrapped over the 2π boundary, and the finite difference is taken over the time dimension. To get back to the time domain waveform, we take the cumulative sum of the instantaneous frequency over the time dimension within each frequency band. This is recombined with the magnitude, and is passed through inverse STFT. This recovers the exact same waveform as the input. The `get_wave_2` function provided in `testing/test_utils.py` in our supplementary shows how we recover the waveform. We thank the reviewer for helping us clarify this point.

---

> > ### Author Response · Authors · 2022-08-02
> > **Response to u3Tz [2/2]**
> >
> >
> > * **[Bitrates of baselines]** We believe you may have misinterpreted our results presented in Table 3. To clarify, our NAFs being smaller than the baselines is an ***advantage***, and is not a sign of NAFs being inferior to the baseline. The Opus and AAC baselines perform worse than NAFs despite being 20x and 40x the size. The code we used for implementing the baselines are in `baselines/make_data_aac.py` and `baselines/make_data_opus.py`, and was provided during the initial submission. We have also detail the version of the encoders we use in section F of our supplementary. We used libopus 1.3.1 and ffmpeg 5.0 native aac as the respective encoders. libopus was set to use maximum complexity for best quality, use music mode for better wideband performance, and use constrained variable bitrate mode. aac was set to use constant bitrate mode. In the revision, we will also mention in the main paper where to find the baseline details.
> >
> >
> > > **Q3) Other clarifications**
> > * **[NAFs and the time-frequency domain]** Thank you for pointing this out! We correct this in the revision in L48 and L114.
> > * **[Time domain output]** Yes, L113 should just indicate the time domain waveform.
> > * **[Figure of the shared grid network]** Our intention was to show the shared grid network in Figure 2. of our main paper, since it was the best performing architecture. We will highlight in Figure 2. that we are showing the "shared grid" design, and further include this figure in the supplementary to provide a better comparison.
> > * **[Dataset details]** We discussed the restricted parameterization of SoundSpaces in section 4.2, and note that it is restricted to a 2D plane. In the revision, we will move the specifics about both SoundSpaces and MeshRIR into section 4.1.
> >
> >
> > We thank the reviewer u3Tz for providing detailed and thoughtful feedback. Following their suggestions, we have run an evaluation to measure how well our framework preserves the binaural cues. We would like to highlight that code is provided in our supplementary for reproducibility. We do note that there may have been a misunderstanding regarding the size of our NAFs, and hope that our clarifications will aid the reviewer in their final evaluation, particularly in light of our additional results.
> >
> > [1] Rafaely, Boaz, et al. "Interaural cross correlation in a sound field represented by spherical harmonics." (2010)
> >
> > [2] Andreopoulou, Areti, et ak. "Identification of perceptually relevant methods of inter-aural time difference estimation." (2017)
> >
> > [3] Srinivasan, Pratul P., et al. "Nerv: Neural reflectance and visibility fields for relighting and view synthesis." (2021)
> >
> > [4] Richard, Alexander, et al. "Neural synthesis of binaural speech from mono audio." (2020)
> >
> > [5] Engel, Jesse, et al. "Gansynth: Adversarial neural audio synthesis." (2019).

---

> > > ### Comment · Reviewer_u3Tz · 2022-08-03
> > > **Significant improvements**
> > >
> > > The authors have put significant effort in addressing my main concern, namely, the lack of evidence supporting the claim that the proposed framework is correctly rendering *spatial* binaural audio. The use of the IACC is appropriate and reveals convincing results.
> > >
> > > The authors have also carefully and correctly adressed my remarks concerning the clarity and the precision of the related work and methodological parts.
> > >
> > > Regarding the bitrates: I know that having a lower memory footprint is an advantage for NAF. My comment was just about the apparent contradiction between the statements "The bitrates are chosen to approach the storage costs for our NAF" and "The Opus and AAC baselines perform worse than NAFs despite being 20x and 40x the size". This contradiction needs to be resolve in the final paper, which I trust the authors to do.
> > >
> > > Despite these important issues that are now resolved, this paper constitutes a strong contribution. Therefore, I am willing to change my overall score from 3 to 8

---

### Official Review · Reviewer_jrJA · 2022-07-13

**Rating:** 5
**Confidence:** 4
**Soundness:** 3 good
**Presentation:** 3 good
**Contribution:** 2 fair

**Summary:**

Modeling room and scene acoustics is vital for AR and VR problems. Realistic rendering of spatialized audio is important. This paper builds a representation learning model/schema for parametrizing such acoustic responses via a encoder-decoder style deep network interpolation machine. This is a neat idea. The paper is written well and the benefits of the proposal are explored.

**Questions:**

Apart from addressing the aspects mentioned in Weaknesses, some other points to expand are:
1. Talk a little more clearly about the decoder setup, and how does this architecture relate to recent advances in NNs for signal parameterization of impulse responses (like https://ieeexplore.ieee.org/abstract/document/9746135)
2. The resolution level of T60 is rather coarse to clearly articulate the benefits of the parameterization in the paper. Can we look at the IRs themselves and capture differences in DRR or early reflections? DRR in particular should show clear differences between this proposal and signal processing baselines.

**Limitations:**

Yes. This was addressed.

**Strengths And Weaknesses:**

Strengths:
1. Clearly motivated for NN modeling. Well written and presented.
2. Reasonable to expect the benefits of NeRF to extend to acoustic fields.

Main weakness:
Room and Scene acoustics is a well studied area in Audio. In particular, interpolation of room responses with and without geometry modeling including accounting for diffraction and dispersion patterns around object boundaries is being actively studied in signal processing and acoustics. Although the authors do build the related works, there are still relevant baselines that are missing (both from discussions and from evaluations perspective) like https://ieeexplore.ieee.org/document/7987742 and others. This is not to say that the evaluations are invalid, but more to understand what the network is really learning compared to classical approaches.
Additionally a bulk of the content in Section 3 introing the acoustic functions can be reduced and cited to appropriate references. Saving much real-estate.

---

> ### Author Response · Authors · 2022-08-02
> **Response to jrJA [1/2]**
>
> We are encouraged by your assessment that modeling scene acoustics is an important question and that our approach is a novel one. We thank Reviewer jrJA for the detailed and constructive review. Below are our responses to specific comments. We look forward to further discussion, and are happy to answer any questions.
>
> > **Q1) Comparison to previous sound field models**
> We agree it is important to highlight the difference between NAFs and past work.
>
> Prior work has proposed both parametric and non-parametric methods to interpolate the sound field. Parametric methods typically only capture perceptually relevant cues, while non-parametric methods seek to estimate the sound field itself. Models have chosen to model the sound field as a linear composition of spherical or plane wave expansions. Methods similar to [1] typically leverage priors or assumptions about the sound field, such as physical constants, far field sound sources, or the position of the receivers. While these assumptions may hold true in certain settings, acoustic environments can be complex and deviate from model priors. Unlike these traditional approaches, our NAFs are learned from data. Furthermore, different from past approaches which typically estimate a sound field for a fixed source, our NAFs enable the arbitrary positioning of both the source and receiver.
>
> Since there is no public implementation of [1], here we provide additional quantitative results using the method described in "Kernel Ridge Regression With Constraint of Helmholtz Equation for Sound Field Interpolation" [2] on the MeshRIR dataset. We use two different variants of the model, the first using their original parameters which include a 500Hz low-pass filter, and the second where we modify the model to use the unfiltered RIR. We use their original proposed regularization value of 0.1.
>
> |                      | Spectral  | T60       | DRR       |
> |----------------------|-----------|-----------|-----------|
> | Ridge-Orig     | 2.539     | 8.192     | 2.497     |
> | Ridge-Unfiltered | 1.370     | 6.294     | 3.702     |
> | NAF (Dual)           | **0.403**     | 4.201     | 0.992     |
> | NAF (Shared)         | **0.403** | **4.191** | **0.972** |
>
> We find our model consistently out performs this baseline. We will include a discussion of [1, 2] and related methods in our updated revision.
>
> > **Q2) Differences in parameterization to "Deep Impulse Responses (DIRs)"**
>
> We want to first clarify that our work is concurrent with [3].
>
> Both [3] and our work parameterize the impulse response as an continuous implicit function. However, DIRs assume a stationary source or receiver, and in practice they focus on a static receiver with emitters distributed on the sphere. NAFs allow both the source and receiver to move freely within a room, but requires us to model a much larger and complex set of impulse responses. This is a fundamentally more challenging problem.
>
> An additional difference is our parameterization of the output. NAFs parameterize the output as log-magnitude and instantaneous frequency (phase) [4], while DIRs output a time domain waveform directly. We experimented with using the representation and MSE training loss as proposed in DIRs, and these results are presented in section **H** of the revised supplementary.
>
> We observed that while outputting the waveform succeeds when modeling a small subset of the impulse responses, the network would only output an over-smoothed waveform when modeling an entire scene. We experimented with increasing the frequency of the fourier features, as this has been suggested to improve the ability of the network to model high frequency data [5]. However we found that this would introduce high frequency noise into the predicted impulse response. This led us to adopt an STFT based output representation. Prior work on using implicit networks for audio representations have similarly modeled either the log-magnitude of the STFT or the full magnitude-phase STFT [6, 7].

---

> > ### Author Response · Authors · 2022-08-02
> > **Response to jrJA [2/2]**
> >
> > > **Q3) Results for the Direct-to-Reverberant Ratio**
> >
> > We agree that Direct-to-Reverberant Ratio (DRR) is a useful metric for characterizing room impulse responses. Here we present the mean absolute error of the DRR for each method:
> >
> > |              | Large 1   | Large 2   | Medium 1  | Medium 2  | Small 1   | Small 2   | MeshRIR   | Mean      |
> > |--------------|-----------|-----------|-----------|-----------|-----------|-----------|-----------|-----------|
> > | AAC-nearest  | 1.748     | 2.424     | 1.344     | 1.343     | 1.213     | 1.108     | 1.286     | 1.495     |
> > | AAC-linear   | 1.797     | 2.147     | 1.457     | 1.458     | 1.117     | 1.226     | 1.222     | 1.490     |
> > | Opus-nearest | 2.931     | 3.275     | 2.756     | 2.769     | 3.548     | 3.255     | 2.698     | 3.033     |
> > | Opus-linear  | 2.645     | 2.771     | 2.381     | 2.370     | 3.266     | 2.882     | 2.529     | 2.692     |
> > | DSP          | 3.559     | 4.421     | 4.727     | 4.805     | 5.622     | 6.723     | N/A       | 4.976     |
> > | NAF (Dual)   |1.645      | 1.830     |1.113      |**1.082**  | **0.796** |**0.799**  | 0.992     |1.179      |
> > | NAF (Shared) | **1.468** | **1.793** | **1.083** | 1.089 | 0.829 | 0.837 | **0.972** | **1.153** |
> > * Mean absolute error of the DRR, units in dB. Lower is better.
> >
> > Note the DSP baseline was not implemented for MeshRIR due to the lack of absolute coordinates.
> >
> >
> > We thank the reviewer for the suggestions, and have added additional quantitative comparisons with a sound field interpolation method alongside DRR results. Following your suggestion, we have also reduced the length of section 3.1 in the revision. We will include additional discussion and add these results to the revision.
> >
> > [1] Antonello, Niccolo, et al. "Room impulse response interpolation using a sparse spatio-temporal representation of the sound field." (2017)
> >
> > [2] Ueno, Natsuki, et al. "Kernel ridge regression with constraint of Helmholtz equation for sound field interpolation." (2018)
> >
> > [3] Richard, Alexander, et al. "Deep Impulse Responses: Estimating and Parameterizing Filters with Deep Networks." (2022)
> >
> > [4] Engel, Jesse, et al. "Gansynth: Adversarial neural audio synthesis." (2019).
> >
> > [5] Tancik, Matthew, et al. "Fourier features let networks learn high frequency functions in low dimensional domains." (2020)
> >
> > [6] Gao, Ruohan, et al. "Objectfolder: A dataset of objects with implicit visual, auditory, and tactile representations." (2021)
> >
> > [7] Du, Yilun, et al. "Learning signal-agnostic manifolds of neural fields." (2021)

---

### Author Response · Authors · 2022-08-02
**General response (Part 1/2)**

We are grateful to all reviewers for their constructive comments which we agree will significantly improve the communication of our work.

We are very encouraged by reviewers’ evaluation on the significance and novelty of this work. All four reviewers find that our work on Neural Acoustic Fields (NAFs) to be novel (“This is a neat idea” (jrJA), “the general idea of NAFs is novel, interesting, and potentially impactful” (u3Tz), “the first, to my knowledge” (e4mi), “method appears to work well” (ossn)).

## 1. General clarifications
### 1.1 Network details and reproducibility
Our method is fully reproducible. We have included a folder of our code, which contains hyperparameters, network architecture, and baselines as part of our supplementary material submitted. We hope the code will help the community reproduce our work and inspire later studies.
### 1.2 Differences from prior work
We would first like to clarify that our work is concurrent with [1].

NAFs differentiates itself by learning a mapping for all possible emitter and listener locations in a scene. This is **fundamentally different** from prior work, which are in practice learned with a non-moving emitter or listener [1,2], or use handcrafted parameterizations of the sound field [3]. We demonstrate that by augmenting the network with shared geometric features that are shared by the emitter and listener, we can achieve a model that is better than a network not using, or using non-shared geometric features. We show that NAFs are significantly more compact than traditional audio coding baselines, and can achieve higher quality when evaluated on T60, spectral, DRR, or IACC error. We further show that audio representations learned by NAFs are informative of scene structure, making it a useful non-visual unsupervised scene representation.

---

> ### Author Response · Authors · 2022-08-02
> **General response (Part 2/2)**
>
> ## 2. Additional Experiments
> The reviewers also suggest that additional metrics and baselines will make the paper stronger, highlight its strengths, clarify potential limitations, and outline directions for future work. We agree, and have augmented our revision with additional qualitative results. We have added an additional baseline, results for direct-to-reverberant ratio (DRR) to better characterize the early components, as well as results for the interaural cross correlation coefficient (IACC) to characterize the spatial cues. We provide these metrics here, and will include them in the revision.
>
> ### 2.1 Interpolation baseline
> Here we compare the method proposed in [4] on the MeshRIR dataset. Where "Constrained-Orig" uses the 500Hz low pass filter as used by the authors, while "Constrained-Unfiltered" is our modification which uses the non-filtered impulse response.
> |                      | Spectral  | T60       | DRR       |
> |----------------------|-----------|-----------|-----------|
> | Constrained-Orig     | 2.539     | 8.192     | 2.497     |
> | Constrained-Unfiltered | 1.370     | 6.294     | 3.702     |
> | NAF (Dual)           | **0.403**     | 4.201     | 0.992     |
> | NAF (Shared)         | **0.403** | **4.191** | **0.972** |
>
> ### 2.2 DRR metric
>
>
> The Direct-to-reverberant ratio is used to measure the ratio of the energy coming from direct region. We find that NAFs have lower DRR error than baseline methods.
>
> |              | Large 1   | Large 2   | Medium 1  | Medium 2  | Small 1   | Small 2   | MeshRIR   | Mean      |
> |--------------|-----------|-----------|-----------|-----------|-----------|-----------|-----------|-----------|
> | AAC-nearest  | 1.748     | 2.424     | 1.344     | 1.343     | 1.213     | 1.108     | 1.286     | 1.495     |
> | AAC-linear   | 1.797     | 2.147     | 1.457     | 1.458     | 1.117     | 1.226     | 1.222     | 1.490     |
> | Opus-nearest | 2.931     | 3.275     | 2.756     | 2.769     | 3.548     | 3.255     | 2.698     | 3.033     |
> | Opus-linear  | 2.645     | 2.771     | 2.381     | 2.370     | 3.266     | 2.882     | 2.529     | 2.692     |
> | DSP          | 3.559     | 4.421     | 4.727     | 4.805     | 5.622     | 6.723     | N/A       | 4.976     |
> | NAF (Dual)   |1.645      | 1.830     |1.113      |**1.082**  | **0.796** |**0.799**  | 0.992     |1.179      |
> | NAF (Shared) | **1.468** | **1.793** | **1.083** | 1.089 | 0.829 | 0.837 | **0.972** | **1.153** |
> ### 2.3 IACC metric
> The interaural cross correlation coefficient is used to measure the spatial localization from impulse responses, and is correlated with localization performance in humans. We find that NAFs achieve the lowest IACC error on average.
>
> |              | Large 1   | Large 2   | Medium 1  | Medium 2  | Small 1   | Small 2   | Mean      |
> |--------------|-----------|-----------|-----------|-----------|-----------|-----------|-----------|
> | AAC-nearest  | 236.8     | 184.2     | 213.7     | 215.3     | 264.8     | 272.5     | 231.2     |
> | AAC-linear   | 212.3     | 156.7     | 185.9     | 187.8     | 245.2     | 265.2     | 208.8     |
> | Opus-nearest | 73.75     | 45.97     | 71.97     | 74.70     | 103.8     | **67.40** | 72.93     |
> | Opus-linear  | 75.56     | 48.32     | 73.38     | 77.33     | 109.2     | 78.10     | 76.98     |
> | DSP          | 460.5     | 446.0     | 430.0     | 430.1     | 443.6     | 446.3     | 442.7     |
> | NAF (Dual)   | 74.01     | 45.94     | 71.89     | 74.70     | 103.8     | **67.40** | 72.96     |
> | NAF (Shared) | **73.68** | **45.90** | **71.52** | **73.58** | **103.6** | **67.40** | **72.62** |
> * Mean absolute difference of IACC (unit in seconds, values here multiplied by 1e6). Lower is better.
>
>
> ## Conclusion
> We thank the reviewers for their careful feedback and additional suggestions for evaluation, which will make the paper significantly stronger.
>
> [1] Richard, Alexander, et al. "Deep Impulse Responses: Estimating and Parameterizing Filters with Deep Networks." (2022)
>
> [2] Richard, Alexander, et al. "Neural synthesis of binaural speech from mono audio." (2020)
>
> [3] Chaitanya, Chakravarty R. Alla, et al. "Directional sources and listeners in interactive sound propagation using reciprocal wave field coding." (2020)
>
> [4] Ueno, Natsuki, et al. "Kernel ridge regression with constraint of Helmholtz equation for sound field interpolation." (2018)

---

### Author Response · Authors · 2022-08-09
**Summary of our response and discussion**

We genuinely thank all reviewers for their constructive comments which have contributed to the improvements in our paper. We sincerely appreciate the positive 8-8-8-5 evaluation from reviewers u3Tz, e4mi, ossn, and jrJA.

Here is a summary of our response.
### Contributions
We would like to first emphasize the contributions of this paper:
* We propose Neural Acoustic Fields, which render the sound for arbitrary emitter and listener positions in a scene. Our NAFs are represented as an implicit function, and outputs the log-magnitude and phase information for a given impulse response.
* We demonstrate that by conditioning the NAFs on a learnable spatial grid of features, we can improve the generalizability of our architecture.
* We show that NAFs can learn geometric structure of a scene that can be useful for downstream tasks.

### Additional experiments
* **[Interpolation baseline]** To address the concern of reviewer jrJA, we add additional experimental results using a "kernel ridge regression" baseline
* **[Additional RIR metric]** To address the concern of reviewer jrJA, we add quantitative results on DRR for our impulse responses.
* **[Spatial audio metric]** To address the concerns of u3Tz, we add an additional evaluation of the interaural cross correlation coefficient for our network and baseline outputs. We show that our network can better preserve spatial cues in the binaural impulse response than the baseline.

### Writing
We thank all reviewers for suggestions regarding our writing and clarity. We believe that the clarifications suggested by the reviewers will improve the communication of our work.
* We provide additional details about our network architecture [jrJA, u3Tz, e4mi], baseline setup [u3Tz, e4mi], and prior work [jrJA, u3Tz].
* We clarify that our NAFs are learned in time-frequency STFT domain, and provide additional details about our phase [u3Tz].
* We have provided additional details about our dataset [u3Tz].

We are deeply grateful to the reviewers for their helpful suggestions, which have helped improve our paper significantly. The additional experiments and clarifications will be reflected in the final version as well.

Best,

Authors

---

### Meta-Review · Area_Chair_ax3S · 2022-08-24

**Recommendation:** Accept
**Confidence:** Certain

**Metareview:**

The authors present Neural Acoustic Fields (NAF), which render sounds for arbitrary emitter and listener positions in a scene. Overall, the reviewers are very positive (8-8-8-5). The authors addressed many of the reviewers' questions about previous related work and rendering spatial binaural audio.

**Award:**

No

---

### Decision · Program_Chairs · 2022-09-14

Accept